# Frequent Occurrence of *NRAS* and *BRAF* Mutations in Human Acral Naevi

**DOI:** 10.3390/cancers11040546

**Published:** 2019-04-16

**Authors:** Philipp Jansen, Ioana Cosgarea, Rajmohan Murali, Inga Möller, Antje Sucker, Cindy Franklin, Annette Paschen, Anne Zaremba, Titus J. Brinker, Ingo Stoffels, Dirk Schadendorf, Joachim Klode, Eva Hadaschik, Klaus G. Griewank

**Affiliations:** 1Department of Dermatology, Venerology and Allergology, University Hospital Essen, 45147 Essen, Germany; inga.moeller@uk-essen.de (I.M.); antje.sucker@uk-essen.de (A.S.); annette.paschen@uk-essen.de (A.P.); anne.zaremba@uk-essen.de (A.Z.); ingo.stoffels@uk-essen.de (I.S.); dirk.schadendorf@uk-essen.de (D.S.); joachim.klode@uk-essen.de (J.K.); eva.hadaschik@uk-essen.de (E.H.); 2West German Cancer Center, University Hospital Essen, 45147 Essen, Germany; 3German Cancer Consortium (DKTK), 69120 Heidelberg, Germany; 4Dermatological Sciences, Institute of Cellular Medicine, Newcastle University, Newcastle upon Tyne NE2 4HH, UK; ioana.cosgarea@newcastle.ac.uk; 5Department of Pathology, Memorial Sloan Kettering Cancer Center, New York, NY 10065, USA; muralir@mskcc.org; 6Department of Dermatology, University of Cologne, 50937 Cologne, Germany; cindy.franklin@uk-koeln.de; 7Department of Dermatology, University Hospital Heidelberg, 69120 Heidelberg, Germany; titus.brinker@nct-heidelberg.de; 8National Center for Tumor Diseases (NCT), German Cancer Research Center (DKFZ), Im Neuenheimer Feld 460, 69120 Heidelberg, Germany; 9Dermatopathologie bei Mainz, Bahnhofstraße 2 b, 55268 Nieder-Olm, Germany

**Keywords:** acral, naevi, melanoma, genetics, dermatology

## Abstract

Acral naevi are benign melanocytic tumors occurring at acral sites. Occasionally they can progress to become malignant tumors (melanomas). The genetics of acral naevi have not been assessed in larger studies. In our study, a large cohort of 130 acral naevi was screened for gene mutations known to be important in other naevi and melanoma subtypes by targeted next-generation sequencing. Mutation status was correlated with clinicopathological parameters. Frequent mutations in genes activating the MAP kinase pathway were identified, including *n* = 87 (67%) *BRAF*, *n* = 24 (18%) *NRAS*, and one (1%) *MAP2K1* mutations. *BRAF* mutations were almost exclusively V600E (*n* = 86, 99%) and primarily found in junctional and compound naevi. *NRAS* mutations were either Q61K or Q61R and frequently identified in dermal naevi. Recurrent non-V600E *BRAF*, *KIT*, *NF1*, and *TERT* promoter mutations, present in acral melanoma, were not identified. Our study identifies *BRAF* and *NRAS* mutations as the primary pathogenic event in acral naevi, however, distributed differently to those in non-acral naevi. The mutational profile of acral naevi is distinct from acral melanoma, which may be of diagnostic value in distinguishing these entities.

## 1. Introduction

Naevi are benign proliferations of melanocytes that usually appear in the first decades of life, but may also be present at birth (congenital naevi). The most frequent naevi arising on the skin are designated common acquired naevi. Naevi arising in acral sites can be located on the dorsal or volar aspect of the hands and feet. Acral naevi are generally acquired naevi. Whereas conventional naevi are more common in fair-skinned individuals and associated with UV exposure [1], acral naevi are more frequent in darker skinned individuals, the association with UV exposure less clear [2]. Both pigmented lesions, lentigines have a modest melanocyte hyperplasia, whereas nevi demonstrate melanocytic nest formation [3]. Naevi can progress to melanoma by acquiring additional gene alterations [4,5,6].

Histologically, naevi can be categorized into subtypes, depending on whether they are confined to the epidermis (junctional naevi), the dermis (dermal naevi), or show both epidermal and dermal components (compound naevi). Acral naevi differ somewhat histologically from their more common non-acral cutaneous counterparts, and may demonstrate factors such as asymmetry, irregular pigmentation, or nuclear hyperchromasia [7,8]. In particular, the presence of melanocytes above the basal layer is common in acral naevi and not necessarily associated with malignant behavior, as is the case in melanocytic tumors of other sites. Such factors can make it difficult to histologically distinguish benign from malignant melanocytic proliferations at acral sites, bearing the risk of misdiagnosis [7,8,9,10,11].

Most naevi are epidermal-derived common acquired naevi, harboring activating *BRAF* V600E mutations (80–90%) [12]. In contrast to acquired naevi, congenital naevi, arising in utero or shortly after birth, primarily harbor *NRAS* mutations (approximately 80% of cases [13]). Other less common naevi have signature mutation profiles. Spitz naevi frequently harbor translocations, i.e., in *ALK*, *NTRK1*, *ROS1*, *RET*, *NTRK3*, *MET*, and *BRAF* [14,15,16,17,18,19,20] or *HRAS* mutations and copy number gains [21]. Activating *GNAQ*, *GNA11*, and, less frequently, *CYSLTR2* and *PLCB4* mutations occur in blue naevi [22,23,24]. Blue naevi arising as part of Carney complex can harbor *PRKAR1A* mutations [25], which have also been related to morphologically related pigmented epithelioid melanocytomas [26]. Deep penetrating naevi frequently demonstrate *CTNNB1* alterations in addition to MAPK-activating mutations [27]. In summary, many benign melanocytic proliferations have specific genetic alterations associated with a clinico-pathologic phenotype.

The genetics of melanoma have been well studied [28,29,30,31]. Cutaneous melanoma is genetically classified as *BRAF*-mutant (~50%), *RAS*-mutant (20–30%), *NF1*-mutant (10–15%), or triple wild-type (10–15%). *BRAF* V600 mutations are relevant therapeutically in melanoma, enabling treatment with BRAF and MEK inhibitors [32].

More recent genetic studies have focused on less frequent melanoma subtypes, such as acral melanoma [28,33,34]. They have recognized that acral melanoma has more frequent chromosomal alterations and a lower mutational burden than non-acral cutaneous melanoma [28]. In acral melanoma, *BRAF* and *NRAS* are still the most frequent mutations present (although less frequent than in non-acral cutaneous melanoma), followed by *NF1* and *KIT* mutations. Two studies analyzing cohorts of 21 [33] and 24 [34] acral naevi, both reported frequent *BRAF* mutations.

Understanding the genetics of naevi enables a better understanding of the pathogenesis of these tumors and the genetic underpinnings of their progression to malignant tumors. This knowledge may help design ancillary assays or algorithms predicting the clinical behavior of tumors difficult to classify solely based on histomorphological evaluation.

The goal of our study was to investigate the frequency of activating mutations in a large cohort of acral naevi (130 samples). A custom targeted next-generation sequencing approach was used to assess various genes with high sensitivity, and associations of identified mutations with clinical pathologic parameters were explored.

## 2. Results

### 2.1. Sample Cohort

The study cohort consisted of 130 acral naevus samples from 123 patients (91 females and 32 males) with an average age of 41 years (range 10 to 79). The tumors included 20 (15.4%) junctional, 76 (58.4%) compound, 17 (13.1%) primarily dermal, and 17 (13.1%) dermal naevi. All samples were primary tumors. Available clinical data are listed in Table 1. Additionally, associations of histological type with size and volume of naevi are listed in Appendix A.

### 2.2. Mutation Analysis for Activating Oncogene Driver Mutations

Targeted amplicon sequencing showed that *BRAF* alterations were the most frequent mutations (*n* = 87, 67%), 86 (99%) being c.1799A>T, V600E, and 1 (1%) being a 1799_1801delTGA, V600E/K alteration. *NRAS* Q61 mutations were identified in 24 cases (18%), comprising 13 (54%) c.182A>G Q61R and 11 (46%) c.181C>A Q61K alterations. One *MAP2K1* 305_307delAGA, E102_I103V, c.309_311delCAA, K104del mutation was detected. Recurrent *KIT*, *HRAS*, *KRAS*, or *TERT* promoter mutations were not observed. All identified mutations were mutually exclusive of one another (Figure 1).

### 2.3. Associations of Clinical and Pathological Parameters with Oncogene Mutation Status

An analysis with available clinico-pathological data was performed. Statistically significant associations were found between oncogene mutation status and patient age, with *BRAF*-mutant (mean 40 years) and *NRAS*-mutant (mean 48 years) tumors occurring at older ages than wild-type tumors (mean 35 years; *p* = 0.04). Wild-type naevi were more often junctional than *BRAF*-*/NRAS*-mutant tumors, while *BRAF*-mutant tumors were often compound and *NRAS*-mutant tumors were often dermal (*p* < 0.0001). Complete details are presented in Table 1, and Figure 2 and Figure 3.

## 3. Discussion

Applying a targeted next generation sequencing approach, we analyzed the largest cohort of acral naevi reported to date, identifying highly recurrent *BRAF* and *NRAS* mutations.

The detected *BRAF* mutation frequency of 67% is lower than that described in non-acral cutaneous naevi (78–82%) [12,35]. V600E c.1799T>A alterations accounted for 99% of *BRAF* mutations, with only one exception, a single tumor with a 1799_1801delTGA, V600E/K alteration. This differs from melanoma, in which around 20% of *BRAF* V600 mutations are non-V600E (mostly V600K, but also V600R or others) [28,31,36]. While non-V600E *BRAF* mutations in melanoma were found to be more common in older patients, these mutations consistently represented between 10 and 20% of *BRAF* mutations in the 20–29, 30–39, 40–49, and 50–59 age groups [36].

The 18% *NRAS* mutation frequency we identified in acral naevi is considerably higher than ~6% *NRAS* mutations reported in cutaneous common acquired naevi, of which larger studies identified *NRAS* mutations in around 6% of samples [35,37]. The mutation frequency of Q61K (54%) and Q61R (46%) cannot be reliably compared to existing studies of conventional common naevi, as few *NRAS* mutations (<5 per study) were identified [35,37]. However, the distribution is similar to what has been described in larger melanoma cohorts [28,31,38], where the Q61K and Q61R mutations were the most frequent *NRAS* mutations and distributed relatively equally.

Sequencing the 19 tumors initially found to be *NRAS*- and *BRAF*-wild-type (15%) (applying the primary smaller 16 gene panel), with a larger panel covering 29 genes recurrently mutated in cutaneous or uveal melanoma [39], we only identified one *MAP2K1* mutation (305_307delAGA, E102_I103V, c.309_311delCAA K104del), highly reminiscent of E102_I103del mutations reported to be activating with similar consequences to *BRAF* mutations [40]. In the genes analyzed in our study, no other mutations were identified, making *BRAF* and *NRAS* mutations the most common activating mutations detected in acral naevi.

Tumor suppressor genes, including *CDKN2A* and *TP53*, are not frequently altered in common naevi [41]. In our study, these genes were sequenced in the samples analyzed by the 29-gene panel and no alterations were identified. However, to reliably assess the presence of alterations in these genes, a comprehensive analysis of mutations and copy number alterations would be required.

There was a statistically significant association (*p* < 0.0001) of mutation status with naevi location (Table 1, Figure 2 and Figure 3). *BRAF* mutations were more frequent in superficial melanocytic naevi (junctional and compound). *NRAS* mutations were more frequent in deeper naevi (primarily dermal). In addition, the estimated naevus volume also varied, depending on mutation status (Figure 2). This is probably associated, at least partly, with naevus location, i.e., dermal naevi have a larger volume than junctional naevi (Appendix A). Our study included more acral naevi excised from women. This is probably partially coincidental, however, a predominance for both acral naevi and melanoma in women of different ethnicities has been reported [2,5,42]. The association of increased patient age with the presence of *BRAF* or *NRAS* mutation will need to be confirmed in future studies.

Comparing the mutation profile identified in naevi to published data on acral and cutaneous melanoma, a few findings are intriguing. The *BRAF* mutation frequency of 67% in naevi is far higher than in acral melanoma (around 20%) [28,33,43]. However, the frequency of conventional c.1799C>A V600E mutations in naevi was higher than in melanoma (99% vs. ~80%, respectively [17,22,31]). *NRAS* mutation frequencies were relatively comparable between melanoma and naevi (both approximately 20% [17,22,31]). There are several mutations occurring in acral melanoma that we did not identify in acral naevi, including *NF1*, *KIT*, and *TERT* promoter alterations [28,31,33], recently reported in 15%, 12%, and 11% of acral melanomas, respectively [43]. The strong differences in gene mutations observed suggest that mutation type may be a diagnostic aid in distinguishing histopathologically difficult-to-classify naevi from melanoma. The identification of non-V600E *BRAF*, *KIT*, *NF1*, or *TERT* promoter mutations may warrant a higher level of suspicion for a potentially malignant acral melanocytic tumor.

All of the identified mutations are assumed to have arisen somatically. UV-exposure is expected to play a carcinogenic role, most likely more so in tumors arising on sun-exposed dorsal areas than palmar or plantar areas. However, other mechanisms, such as sporadic non-carcinogen-induced mutations, probably play a greater pathogenic role in these tumors than in non-acral cutaneous naevi.

Melanomas can arise with or without transformation from pre-existing naevi. Most naevi do not transform into melanoma, and many melanomas arise without knowledge of a preexisting naevus [8,44,45]. The genetic findings we have obtained do bring up some interesting aspects. Considering we detected no recurrent non-V600E *BRAF*, *KIT*, or *NF1* mutations in naevi, this may imply that acral melanomas harboring these mutations arise primarily de novo, whereas *BRAF* V600E or *NRAS* Q61 mutant acral melanomas may more frequently arise from pre-existing naevi. Larger studies of acral melanoma with strong epidemiological data would be required to see if such an association does exist.

Our study has some caveats. We analyzed a maximum of 29 genes, and rarer mutations may have been missed. This also applies to translocations, which are rare in acral melanoma [28,43], but may occasionally occur in naevi [15]. Our sequencing panel does not allow copy number alterations to be reliably assessed; however, these have been shown to be very rare in naevi [46]. Strengths of our study are the large number of tumors included (130 naevi) and the use of a next-generation sequencing approach, simultaneously screening for many genes known to be important in melanocytic tumor pathogenesis. This differs from previous studies of naevi, in which Sanger-sequencing of individual gene mutations was performed [12,35,37].

In summary, our study presents the most comprehensive genetic analysis of a large group of acral naevi to date. *BRAF* V600E mutations were most frequent (66%), identified primarily in superficial (junctional and compound) naevi. However, *NRAS* Q61 mutations were also detected (18%), particularly in naevi with a more dermal location. The mutation profile identified in acral naevi differs considerably from acral melanoma, which could prove to be of diagnostic value.

## 4. Materials and Methods

### 4.1. Sample Selection

Samples of acral naevi were obtained from the databases of the Department of Dermatology University Hospital Essen and Dermatopathologie bei Mainz (*n* = 101), Germany. All cases were screened by at least one experienced board-certified dermatopathologist (KGG or EH). The study was done with the approval of the Ethics Committee of the University of Duisburg-Essen, under the IRB-number 18-8426-BO. The study was performed with patient informed consent and conducted in accordance with the Declaration of Helsinki.

### 4.2. Clinical and Histopathological Analysis

Available clinical information was taken from histology reports and patient records. Histological assessment included distinction between junctional, compound, primarily dermal, and dermal naevi. Primarily dermal was defined as naevi having primarily dermal melanocyte nests with up to three epidermal melanocyte nests. The ratio of epidermal to dermal melanocytes had to be >5 to be classified as primarily dermal. Tumors with a lower ratio were classified as compounds. Tumor thickness, tumor width, ratio of epidermal/dermal melanocytes, presence of superbasal intraepidermal melanocytes, signs of dysplasia, fibrosis, pigmentation, and presence of lymphocytes were assessed. The size of naevi was estimated by measuring the maximum width and height in mm and multiplying these to estimate surface area. Estimated naevus volume was calculated as 1/2 × 4/3 × Pi × r^3^.

### 4.3. DNA Isolation

DNA was isolated from 10 µm thick sections, cut from formalin-fixed, paraffin-embedded tumor tissues. The sections were deparaffinised and manually macrodissected. DNA isolation was performed with the QIAamp DNA Mini Kit (Qiagen, Hilden, Germany), according to the manufacturer’s instructions.

### 4.4. Targeted Sequencing

A custom amplicon-based sequencing panel covering 16 genes (Appendix A) was designed and prepared, applying the GeneRead Library Prep Kit from QIAGEN^®^ (19300 Germantown Rd, Germantown, MD 20874, USA), according to the manufacturer’s instructions. Tumors not demonstrating a mutation in the 16 genes were further sequenced by a 29-gene panel (Appendix A) [47].

Adapter ligation and barcoding of individual samples were done, applying the NEBNext Ultra DNA Library Prep Mastermix Set and NEBNext Multiplex Oligos for Illumina from New England Biolabs. Up to 60 samples were sequenced in parallel on an Illumina MiSeq next-generation sequencer. Sequencing analysis was performed, applying the CLC Cancer Research Workbench from QIAGEN^®^, as described [47]. In brief, the following steps were applied. The workflow in CLC included adapter trimming and read pair merging before mapping to the human reference genome (hg19). Insertions and deletions, as well as single nucleotide variant detection, local realignment, and primer trimming, followed. Additional information was obtained regarding potential mutation type, known single nucleotide polymorphisms, and conservation scores by cross-referencing varying databases (COSMIC [48], ClinVar [49], dbSNP [50], 1000 Genomes Project [51] and HAPMAP [52]). The resulting CSV files were further analyzed manually. Mutations affecting the protein coding portion of the gene were considered if predicted to result in non-synonymous amino acid changes. Mutations were reported if the overall coverage of the mutation site was ≥30 reads, ≥5 reads reported the mutated variant, and the frequency of mutated reads was ≥1%. Access to raw sequencing data will be granted upon request.

### 4.5. Associations of Oncogene Mutation Status with Clinical and Pathologic Parameters

We investigated associations of mutation status with available clinical and pathological parameters, using chi-squared tests and Fisher exact tests, as appropriate. All statistical analyses were performed using IBM SPSS Statistics software (version 25.0; International Business Machines Corp., Armonk, NY, USA). A *p*-value of *p* ≤ 0.05 was considered statistically significant.

## 5. Conclusions

Acral naevi demonstrate frequent *BRAF* V600E mutations (66%), primarily in superficial (junctional and compound) locations, and frequent *NRAS* Q61 mutations (18%), particularly in more dermal localized tumors. Differences in mutation profile between acral naevi and melanoma may prove to be of diagnostic value.

## Figures and Tables

**Figure 1 cancers-11-00546-f001:**
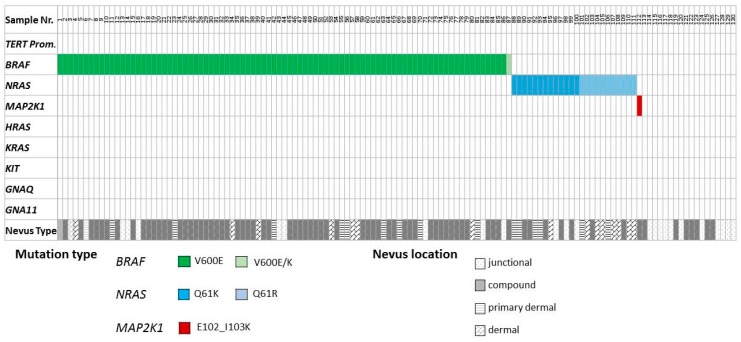
Distribution of activating mutations identified in acral naevi. Distribution of activating mutations identified in different oncogenes in the acral naevus cohort. The resulting amino acid changes as well as naevus location (junctional, compound, primarily dermal, or dermal) are color-coded according to the scheme underneath the illustration.

**Figure 2 cancers-11-00546-f002:**
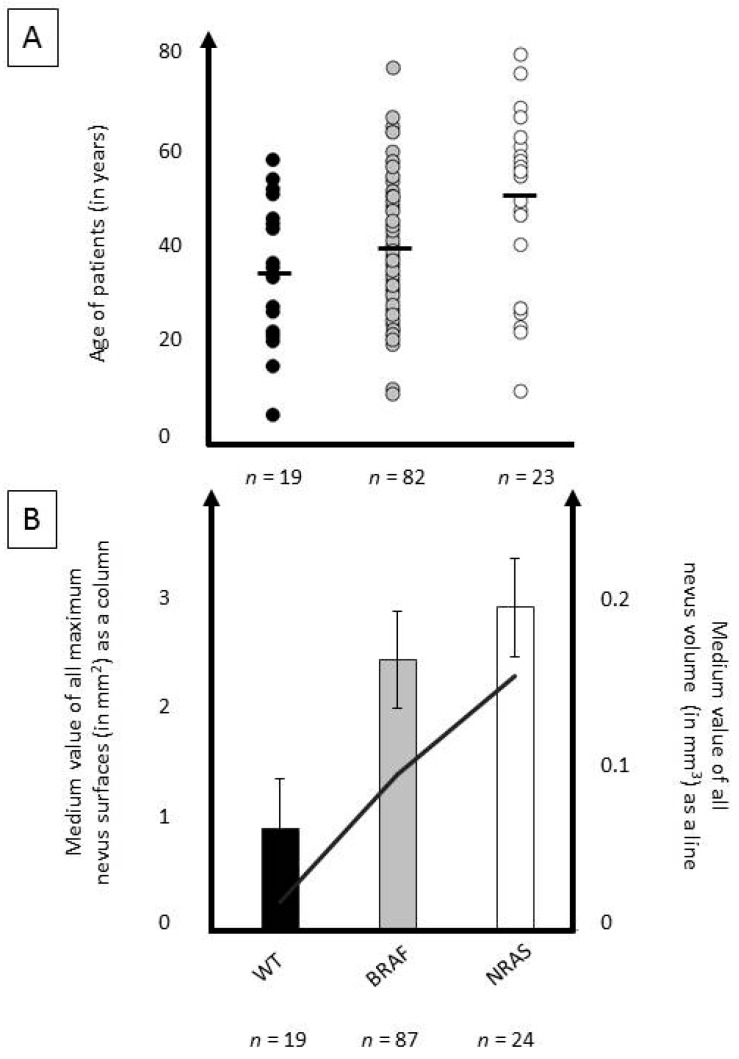
Associations of patients’ age, size, and volume of naevi with oncogene mutation status. (**A**) Distribution of patients according to their age in wild-type cohort (black), *BRAF*-mutant cohort (grey) and *NRAS*-mutant cohort (white) (mean age of cohort indicated by black line). (**B**) Medium size of naevus surface (in mm^2^) depicted as columns for wild-type cohort (in black), *BRAF*-mutant cohort (in grey) and *NRAS*-mutant cohort (in white) with standard deviation. Medium volume of naevi for all three cohorts (WT, *BRAF*, *NRAS*) is depicted as a black line.

**Figure 3 cancers-11-00546-f003:**
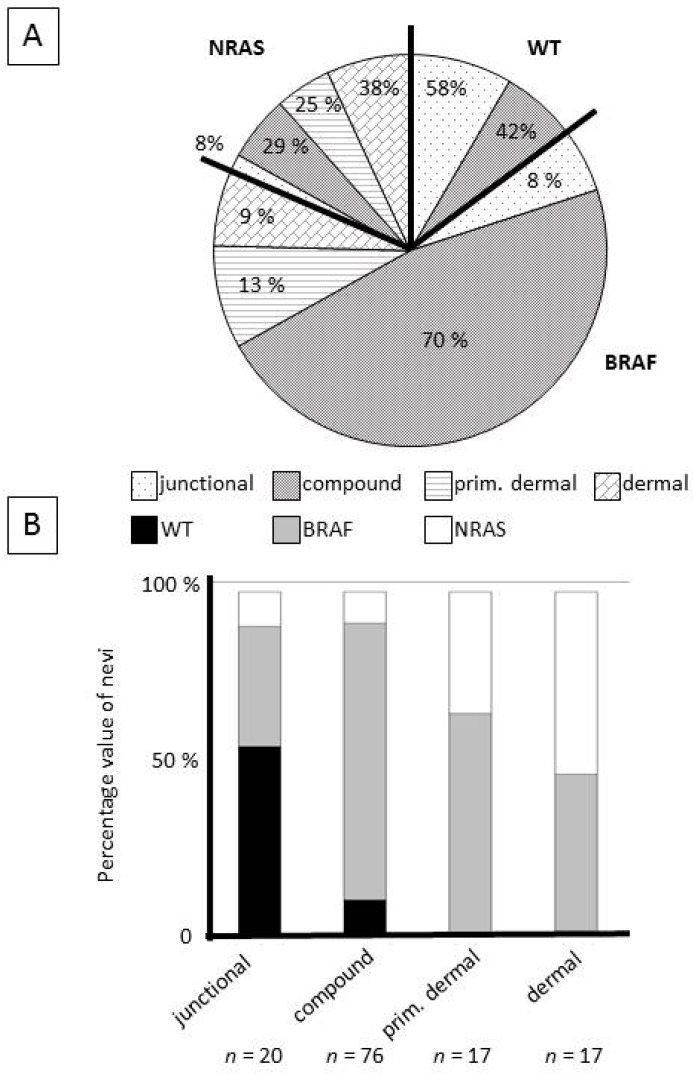
Associations of histological type and oncogene mutation status of naevi. (**A**) Percentage value of different histological naevus types in wild-type (WT), *BRAF*-, and *NRAS*-mutant cohorts. (**B**) Percentage value of wild-type (WT), *BRAF*-, and *NRAS*-mutant naevi in the four different naevus locations (junctional, compound, primarily dermal, and dermal).

**Table 1 cancers-11-00546-t001:** Associations of *BRAF* and *NRAS* mutation status with clinical and pathological parameters.

Variable	Specific Variable	All *n* = 130	%	WT *n* = 19	%	*BRAF*-Mutant *n* = 87	%	*NRAS*-Mutant *n* = 24	%	*p*-Value *
Mean Age (Years)		41		35		40		48		0.04
Sex ^+^	Female	91	74%	14 ^†^	73.7%	60	73.2 %	17	73.9%	0.08
	Male	32	26%	5	26.3%	22	27.5%	6	26.1%	
Sites of Involvement	foot	121	93.1%	16 ^†^	84.2%	81	93.1%	24	100%	0.34
	hand	7	5.4%	2	10.5%	5	5.7%	0	0	
	LND	2	1.5%	1	5.3%	1	1.2%	0	0	-
Sites of Involvement	volar	41	31.6%	5	26.4%	29	33.3%	7	29.2%	0.11
	dorsal	25	19.2%	7 ^†^	36.8%	16	18.4%	2	8.3%	
	LND	64	49.2%	7	36.8%	42	48.3%	15	62.5%	-
Histotype	junctional	20	15.4%	11 ^†^	57.9%	7	8.1%	2	8.3%	<0.0001
	compound	76	58.4%	8	52.1%	61	70.1%	7	29.2%	
	prim. dermal	17	13.1%	0	0	11	12.6%	6	25%	
	dermal	17	13.1%	0	0	8	9.2%	9	37.5%	

WT = wild-type for *BRAF* and *NRAS;* prim. = primarily; * age Kruskal–Wallis test; all others: Fisher exact test; LND: Localization not determined; ^†^ One patient having a *MAP2K1* mutation (was included in the wild-type group); + one woman had three *BRAF*-mutant naevi, one woman had two *BRAF*-mutant naevi, one woman showed two *NRAS*-mutant naevi, two men had two *BRAF*-mutant naevi, and one man showed one *BRAF*-mutant naevus and one WT naevus (this man is represented in the “*BRAF*-mutant” column and in the “WT” column but counted once in the “All” column).

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
