# Peer review of "Frequent Occurrence of NRAS and BRAF Mutations in Human Acral Naevi"

_cancers, 2019, doi:10.3390/cancers11040546_

Reviewer 1 Report

Acral naevi are benign melanocytic tumors occurring at acral sites.  Occasionally they can progress to malignant melanomas.  The genetics of acral naevi have not been evaluated in larger studies.  Thus, the authors conducted a large cohort of 130 acral naevi screened for gene mutations that have been characterized to be important in melanomas by next-generation sequencing.  Mutation status was correlated with clinicopathological parameters.  Frequent mutations in genes activating the MAP kinase pathway were identified, including (67%) BRAF, (18%) NRAS and one (1%) MAP2K1 mutations in acral naevi.  BRAF mutations were almost exclusively V600E (99%) and primarily found in junctional and compound naevi.  NRAS mutations were either Q61K or Q61R and frequently identified in dermal naevi.  Recurrent non-V600E BRAF, KIT, NF1 and TERT promoter mutations, present in acral melanomas, were not identified.  Their study identifies BRAF and NRAS mutations being the primary pathogenic event in acral naevi, distributed differently than in non-acral naevi.  The mutational profile of acral naevi is distinct from acral melanomas, which may be of diagnostic value in distinguishing these entities.

This is a comprehensive study with human clinical samples of acral naevi for MAK kinase-related  genes frequently mutated in human melanomas.  They found that B-RAF mutations and N-RAS mutations were most common in benign acral disease.  I noticed some issues in this manuscript that should be corrected before circulation.

Major Issues.

1) Table 1.  It seems they collected more acral nevi samples from females than males.  Is the disease more common in females than males? 

They should also mention about the influences of sun exposure, and human races (i.e. is the disease more common in African Americans than white people) for the development of acral naevi.

They should also comment on the difference between naevi and lentigo.

2) Table 1.  I found two lines for dermal in histotype.  Both 13.1%, with differences in the frequencies of B-RAF mutation and N-RAS mutation.  Why are there two lines?

3) So what are the differences between benign naevi and malignant melanomas?  I understood the difference in the frequency of mutations and the involvement of other genes like NF1, KIT, and TERT.  One most important things that should be analyzed and mentioned in this manuscript is the alterations of INK4a/ARF and p53.  Probably these genes are mutated or silenced in malignant melanomas which are much more invasive/metastatic than naevi.  Do they have the data for these?  Why not mentioned in the manuscript?

4) Results.  They comment on the mutations on the telomerase promoter.  The mutations were not observed in acral naevi.  How about overexpression of TERT by quantitative RT-PCR?  It should be common in human cancers.

5) So what are the messages to clinicians?  Should acral naevi be treated with surgery only?  Should clinicians use B-RAF inhibitors and N-RAS inhibitors when they find naevi patients?

I leaned that approaches to directly target mutant RAS have not been successful in cancer therapy, and thus most efforts have focused on indirect approaches to block RAS membrane association or downstream effector signaling.

Author Response

We would like to thank the reviewers for their additional positive feedback and constructive suggestions. Below is a point-by-point response to each of the comments.

Point by point response for Reviewer 1:

Acral naevi are benign melanocytic tumors occurring at acral sites.  Occasionally they can progress to malignant melanomas.  The genetics of acral naevi have not been evaluated in larger studies.  Thus, the authors conducted a large cohort of 130 acral naevi screened for gene mutations that have been characterized to be important in melanomas by next-generation sequencing.  Mutation status was correlated with clinicopathological parameters.  Frequent mutations in genes activating the MAP kinase pathway were identified, including (67%) BRAF, (18%) NRAS and one (1%) MAP2K1 mutations in acral naevi.  BRAF mutations were almost exclusively V600E (99%) and primarily found in junctional and compound naevi.  NRAS mutations were either Q61K or Q61R and frequently identified in dermal naevi.  Recurrent non-V600E BRAF, KIT, NF1 and TERT promoter mutations, present in acral melanomas, were not identified.  Their study identifies BRAF and NRAS mutations being the primary pathogenic event in acral naevi, distributed differently than in non-acral naevi.  The mutational profile of acral naevi is distinct from acral melanomas, which may be of diagnostic value in distinguishing these entities.

This is a comprehensive study with human clinical samples of acral naevi for MAK kinase-related genes frequently mutated in human melanomas.  They found that B-RAF mutations and N-RAS mutations were most common in benign acral disease.  I noticed some issues in this manuscript that should be corrected before circulation.

Authors´ response: We appreciate the reviewer´s positive assessment of our work. The individual suggestions to improve the manuscript brought up in the review are addressed below.

1) Table 1.  It seems they collected more acral nevi samples from females than males.  Is the disease more common in females than males?

Authors´ response: This is a good point. We believe this is primarily coincidental, however a predominance of acral naevi and melanoma in females has been reported by a number of studies. We have briefly discussed this in the manuscript. It now reads:

“Our study included more acral naevi excised from women. This is probably partially coincidental, however a predominance for both acral naevi and melanoma in women of different ethnicities has been reported [2,5,42].

They should also mention about the influences of sun exposure, and human races (i.e. is the disease more common in African Americans than white people) for the development of acral naevi.

They should also comment on the difference between naevi and lentigo.

Authors´ response: We have addressed this in the introduction, it now reads as follows:

“Whereas conventional naevi are more common in fair-skinned individuals and associated with UV-exposure [1], acral naevi are more frequent in darker skinned individuals, the association with UV-exposure less clear [2]. Both pigmented, lentigines have a modest melanocyte hyperplasia, nevi demonstrate melanocytic nest formation [3].”

2) Table 1.  I found two lines for dermal in histotype.  Both 13.1%, with differences in the frequencies of B-RAF mutation and N-RAS mutation.  Why are there two lines?

Authors´ response: We appreciate the reviewer for bringing this up. One of the lines is “primarily dermal” the other “dermal”. We have now adapted Table 1 to make this differentiation clearer.

3) So what are the differences between benign naevi and malignant melanomas?  I understood the difference in the frequency of mutations and the involvement of other genes like NF1, KIT, and TERT.  One most important things that should be analyzed and mentioned in this manuscript is the alterations of INK4a/ARF and p53.  Probably these genes are mutated or silenced in malignant melanomas which are much more invasive/metastatic than naevi.  Do they have the data for these?  Why not mentioned in the manuscript?

Authors´ response: We thank the reviewer for this suggestion. INK4a/ARF and p53 (TP53) were not covered in our smaller sequencing panel. Both are included in the larger 29 gene sequencing panel we applied to samples where no mutation was observed when sequenced by the smaller gene panel. In none of these cases where the 29 gene panel was applied were mutations in CDKN2A or TP53 observed. We have included this information and discussed it in a short paragraph in the discussion section. It now reads:

“Tumour suppressor genes, including CDKN2A and TP53 are not frequently altered in common naevi [41]. In our study, these genes were sequenced in the samples analyzed by the 29-gene panel and no alterations were identified. However, to reliably assess the presence of alterations in these genes, a comprehensive analysis of mutations and copy number alterations would be required.”

4) Results.  They comment on the mutations on the telomerase promoter.  The mutations were not observed in acral naevi.  How about overexpression of TERT by quantitative RT-PCR?  It should be common in human cancers.

Authors´ response: Most melanomas (like many other malignancies) have TERT promoter mutations, so sequencing for mutations is the most commonly performed analysis for TERT dysregulation. Performing RT-PCR could be informative, but it is unlikely that significant expression alterations will be found (given that naevi are benign neoplasms, and TERT promoter mutations have not been reported in benign neoplasms with any great frequency). In any event, qRT-PCR analysis is beyond the scope of the current study.

5) So what are the messages to clinicians?  Should acral naevi be treated with surgery only?  Should clinicians use B-RAF inhibitors and N-RAS inhibitors when they find naevi patients? I leaned that approaches to directly target mutant RAS have not been successful in cancer therapy, and thus most efforts have focused on indirect approaches to block RAS membrane association or downstream effector signaling.

Authors´ response: The take home message for the clinician is that in acral melanocytic neoplasms whose dignity is difficult to determine based solely on histological analysis, genetic differences can potentially aid in supplying a correct diagnosis. RAS signaling is (to our knowledge) still very difficult to target, however in the tumours we assessed, inhibitor therapies are not necessary.

Reviewer 2 Report

Although the study was performed by employing a large number of samples, it brings not much novelty in the field of sequencing data. The major limitation, also stated by the Authors in the discussion, is that a maximum of 29 genes were analyzed, and they were chosen based on previous studies. Therefore, the manuscript is rather descriptive and hypothesis generating rather than dissolving any scientific problems or discovering novel genetic variants.

I have several specific concerns:

Some statements lack references, eg. in the introduction.

line 71 - the same paper is cited twice.

quality of the figures is very low, they are not readable.

line 147 - that conclusion is too speculative and not supported by the limited data.

Are raw sequencing data available in public databases

Author Response

We would like to thank the reviewers for their additional positive feedback and constructive suggestions. Below is a point-by-point response to each of the comments.

Point by Point response for Reviewer 2:

Although the study was performed by employing a large number of samples, it brings not much novelty in the field of sequencing data. The major limitation, also stated by the Authors in the discussion, is that a maximum of 29 genes were analyzed, and they were chosen based on previous studies. Therefore, the manuscript is rather descriptive and hypothesis generating rather than dissolving any scientific problems or discovering novel genetic variants. I have several specific concerns:

Authors´ response: We have attempted to address the reviewer´s comments as best possible below.

1) Some statements lack references, eg. in the introduction and line 71 - the same paper is cited twice.

Authors´ response: We appreciate the reviewer picking this up and have included references in the introduction and corrected the citations in line 71.

2) quality of the figures is very low, they are not readable.

Authors´ response: We apologize for the figure quality. The journal requested the figures are included in the manuscript text, which significantly decreased the quality of the figures. We have now done our best to increase the figure quality in the manuscript and have attached the figures again as an independent file, in which the quality should be high.

3) line 147 - that conclusion is too speculative and not supported by the limited data.

Authors´ response: We have modified the statement to address solely the findings of our study. It now reads:

“In the genes analyzed in our study, no other mutations were identified, making BRAF and NRAS mutations the most common activating mutations detected in acral naevi.”

4) Are raw sequencing data available in public databases

Authors´ response: The sequencing data will be made available for download upon publication (GEO - Gene Expression Omnibus)

Reviewer 3 Report

Comments on manuscript "Frequent Occurrence of NRAS and BRAF Mutations 2 in Human Acral Naevi".

The authors have investigated 130 acral nevi and performed sequencing of 16 candidate genes. In case no BRAF or NRAS mutation was found an additional set of 29 genes were sequenced.

From the 130 samples 87 were mutated in BRAF, 24 in NRAS and 1 in MAP2K1.

Currently the COSMIC database only features two samples with mutation status for acral nevi. This indicates that more data on acral nevi is required. The current manuscript tries to fill this gap and contributes to characterise the difference between nevi and malignant melanoma.

The manuscript is well structured and written in excellent English.

Issues to address:

Are all mutations detected classified as somatic mutations? What is the view of the authors how the identified mutations have arisen?

Why was GRIN2A, a common mutated gene in melanoma, not included in the sequencing list?

Although nevi contain mutations also found in melanoma, nevi seldomly progress to malignancy. Authors should comment on this phenomenon and present their views.

Figure2, Figure3, FigureS1 are of poor resolution. Readability is not good and needs to be improved.

Figure 3B: Colour for WT and BRAF samples is too similar and can hardly be distinguished. Different gray scaling or colour should be used.

Author Response

We would like to thank the reviewers for their additional positive feedback and constructive suggestions. Below is a point-by-point response to each of the comments.

Point to Point response for Reviewer 3:

The authors have investigated 130 acral nevi and performed sequencing of 16 candidate genes. In case no BRAF or NRAS mutation was found an additional set of 29 genes were sequenced.

From the 130 samples 87 were mutated in BRAF, 24 in NRAS and 1 in MAP2K1.

Currently the COSMIC database only features two samples with mutation status for acral nevi. This indicates that more data on acral nevi is required. The current manuscript tries to fill this gap and contributes to characterise the difference between nevi and malignant melanoma.

The manuscript is well structured and written in excellent English.

Authors´ response: We appreciate the reviewers´ positive assessment of our manuscript and have attempted to address the comments as best possible below.

1) Are all mutations detected classified as somatic mutations? What is the view of the authors how the identified mutations have arisen?

Authors´ response: This is a good point brought up by the reviewer. We are certain that the mutations are somatically acquired. In terms of pathogenesis, UV-exposure is almost certainly still a relevant factor, at least for naevi arising on the dorsal aspect of the hand and foot and to some extent for naevi on the plantar and palmar surface, however, other mechanisms are likely more relevant. Sporadic non-carcinogen-induced mutations likely play a relevant role in many cases. It now reads:

“All of the identified mutations are assumed to have arisen somatically. UV-exposure is expected to play a carcinogenic role, most likely more so in tumours arising in sun exposed dorsal areas than palmar or plantar areas. However, other mechanisms, including sporadic non-carcinogen-induced mutations probably play a greater pathogenic role in these tumours than in non-acral cutaneous naevi.”

2) Why was GRIN2A, a common mutated gene in melanoma, not included in the sequencing list?

Authors´ response: To our knowledge, GRIN2A has not been reported to be mutated in naevi. It is of course speculative, but we believe it is very unlikely it to be a relevant oncogene in our cohort. This question could be addressed in future studies.

3) Although nevi contain mutations also found in melanoma, nevi seldomly progress to malignancy. Authors should comment on this phenomenon and present their views.

Authors´ response: We appreciate the reviewer bringing this up. This is of course a highly relevant point clinically. We have now entered a paragraph addressing this in the discussion section. It now reads:

“Melanomas can arise with or without transformation from preexisting naevi. Most naevi do not transform into melanoma, and many melanomas arise without knowledge of a preexisting naevus. The genetic findings we have obtained do bring up some interesting aspects. Considering we detected no recurrent non-V600E BRAF, KIT or NF1 mutations in naevi, this may imply that acral melanomas harboring these mutations arise primarily de novo, whereas BRAF V600E or NRAS Q61 mutant acral melanomas may more frequently arise from pre-existing naevi. Larger studies of acral melanoma with strong epidemiological data would be required to see if such an association does exist.”

4) Figure2, Figure3, FigureS1 are of poor resolution. Readability is not good and needs to be improved. Figure 3B: Colour for WT and BRAF samples is too similar and can hardly be distinguished. Different gray scaling or colour should be used.

Authors´ response: For the figure quality, we apologize, this is based on the journal requesting figures are incorporated in the manuscript (word) file. We have now done our best to improve the figure quality in the manuscript and have added all figures as a separate file, which should be of high quality. The color and grey scale have been altered as requested.

Round  2

Reviewer 2 Report

The comments and concerns have been addressed, however, how the Authors want to include GEO accession number in the manuscript when they state that "The sequencing data will be made available for download upon publication (GEO - Gene Expression Omnibus)"? It is possible to upload sequencing data, gain accession number and put it in the materials and methods while scheduling the public release of the raw data on the specific date (eg. when the manuscript is accepted for publication).